# Ultrasound Evaluation of Extracranial Cerebral Circulation (The Common, External and Internal Carotid Artery) in Different Breeds of Dogs

**DOI:** 10.3390/ani13101584

**Published:** 2023-05-09

**Authors:** Marieta K. Ševčíková, Mária Figurová, Karol Ševčík, Marián Hluchý, Michal Domaniža, Mária Lapšanská, Zuzana Drahovská, Zdeněk Žert

**Affiliations:** 1Small Animal Clinic, University of Veterinary Medicine and Pharmacy in Košice, Komenského 73, 041 81 Košice, Slovakia; marietakurillova@gmail.com (M.K.Š.); sevcik.karol@gmail.com (K.Š.); marian.hluchy@uvlf.sk (M.H.); domaniza.michal@gmail.com (M.D.); maria.lapsanska@uvlf.sk (M.L.); zuzana.drahovska@uvlf.sk (Z.D.); 2Clinic of Horses, University of Veterinary Medicine and Pharmacy in Košice, Komenského 73, 041 81 Košice, Slovakia

**Keywords:** Doppler ultrasound, carotid arteries, dog, velocity, resistive index, weight

## Abstract

**Simple Summary:**

Doppler ultrasonography has become increasingly available in veterinary medicine and is the gold standard method for evaluating the heart and vasculature in dogs. There is an information gap in ultrasound evaluation of the carotid arteries with respect to breed variations. This study aimed to investigate the quantitative and qualitative parameters of blood flow in the carotid arteries and to describe their correlations or associations with the breed, sex, and weight of the examined dogs. The study included eight breeds—104 dogs—and found that the highest variation in the spectral waveform and velocities of blood flow between breeds occurred in the common carotid artery. Knowledge of the physiological signs of Doppler spectral waveforms is important due to their high specificity for each evaluated vessel. Our results might help with clinical identification of pathologies based on spectral flow patterns. Every practitioner performing clinical and/or Doppler ultrasound examination of a patient should be able to distinguish physiological from pathological findings. The present study supports the hypothesis that very good reliability was obtained by pre-examination training and by following a precise methodology.

**Abstract:**

Noninvasive Doppler ultrasonography (US) examination is a standard method for the clinical evaluation of the carotid arteries. Extracranial cerebral circulation includes the common carotid artery (CCA), the external carotid artery (ECA), and the extracranial part of the internal carotid artery (ICA). The present study was conducted with the objective of assessing physiological normative values and describing the appearance of spectral waveforms of extracranial arteries in 104 healthy dogs of eight breeds divided into four groups according to weight. We also focused on searching for correlations for carotid blood velocities with the resistive index (RI), body weight and diameter, and differences between observers and the influence of sex in the calculations of Doppler parameters. In the evaluated breeds, significant differences were found in the velocity of peak systolic velocity (PSV) and end diastolic velocity (EDV). There was a strong correlation between body weight and peak systolic velocity, the RI index and CCA diameter. The intra-observer agreement for the PSV and EDV parameters in each vessel was considered excellent reliability, and overall interobserver agreement showed very good reliability. This study could improve the descriptions of physiological values and waveforms recorded in carotid arteries. Defining the physiological values of velocity and the resistive index (RI) helps in the easier identification of pathology and diagnosis of disease. Our results may be used for further studies of vascular diseases in veterinary medicine that correlate with the pathology of neurological disorders of ischemic origin, further in thromboembolism, oncologic disease and degenerative, proliferative and inflammatory diseases of the arteries that lead to stenosis

## 1. Introduction

Doppler ultrasonography (US) is an effective and useful tool in veterinary medicine imaging due to its accessibility, low price, lack of radiation exposure or sedative intake, and the possibility to perform a bedside exam even in critically ill patients [1,2,3,4,5,6,7,8]. Waveform spectrum analysis reveals blood flow information, including physiological details and pathological processes [9,10,11,12,13].

US evaluation of the common carotid artery (CCA), external carotid artery (ECA), and extracranial part of internal carotid artery (ICA) is best seen by varying the scanning angle between a dorsal and parasagittal plane. The extracranial part of the ICA arises at an approximately 30-degree angle from the ECA and courses internally. In some dogs, the carotid sinus is seen as a focal dilation at the origin of the ICA (Figure 1). Assessing the intracranial part of the ICA is possible with the transcranial Doppler technique.

The ECA continues in a straight line rostrally from the common carotid artery (CCA) [6].

Vascular diseases, such as thrombosis, atherosclerosis, infarction, vasculitis, neoplastic infiltration, and stenosis, are reported in veterinary practice [14,15,16]. Additionally, cognitive injury in human patients with neurologic deficits arising from carotid stenosis is known [17]. Diagnosing the mentioned indications through US examination should become regular practice in the monitoring of dogs with neurological disorders and deficits by stroke, brain ischemia, dementia, or oncologic disease [1,18,19].

Each artery has a characteristic appearance, a specific flow pattern, as well as typical spectral waveforms that indicate a certain pathology [3,4,11,12,20,21,22,23]. Features of spectral Doppler waveforms of the CCA, ECA, and ICA involve major descriptors, such as phasicity, resistance, and direction of flow [1,3,4,11,21,24]. Correct interpretation of color and spectral Doppler findings may be influenced by operator-dependent errors and incorrect technical parameters settings. The latter affect the accuracy of carotid arteries values, including the Doppler angle, sample volume box, color Doppler sampling window, color velocity scale, and color gain [17,25,26,27,28,29,30]. The most common source of operator error is misalignment of the spectral Doppler efficiency angle with the direction of blood flow [17,27,28,29,31].

Resistance has been described as arterial spectral Doppler waveforms demonstrating high, intermediate, or low resistance [3,11,24,32]. Based on human reports, the interpretation of changes in Doppler waveforms in arteries is largely difficult due to overlapping and conflicting terminologies [11,31]. Data from CCA investigations have been published [1,4], but records from US evaluation of the ECA and ICA in veterinary medicine have not yet been published.

This article provides a narrative review of the physiological normative values and the spectral waveforms in different breeds divided into four groups according to their size and weight. In this study, we aimed (1) to describe the physiological spectral waveform appearance of the CCA, ECA and ICA and its variations in different groups of breeds and between gender, (2) to compare the parameters peak systolic velocity (PSV), end diastolic velocity (EDV) and resistive index (RI), (3) to determine the relationship of morphometric parameters, such as weight and diameter of the CCA, with hemodynamic parameters, (4) and finally to evaluate intra-observer and inter-observer variability in the protocols developed for spectral waveform measurements.

We hypothesized that there would be (1) a different morphological flow pattern of the spectral waveform between groups of breeds, (2) significantly different blood velocity in the large and giant breeds versus medium and small breeds, (3) a relationship between weight and diameter of the vessel and velocity parameters, (4) and gender-related differences in all parameters.

## 2. Material and Method

### 2.1. Study Population

Eight breeds with 13 dogs in each breed group aged between 1–5 years of age were included and divided based on weight and body size into the four groups: Small breeds (Dachshund, Miniature Schnauzer), medium breeds (Slovakian Hound, Bavarian Mountain Scent Hound), large breeds (Hungarian Short-Haired Pointer, Standard Poodle) and giant breeds (Borzoi and Great Dane). After arrival at the clinic, the dogs were allowed to acclimatize and get used to the new environment for 10–15 min. They were then examined without sedation, and before enrollment, the dogs were deemed healthy based on the results of the inclusion criteria, which were: Without current medication and history of systemic disease, physiological examination within the physiological values.

### 2.2. Spectral Doppler Examination

All examinations were performed using an Aloka Profound Alpha 6 equipped with a linear (5–16 MHz) probe bilaterally for the CCA, ECA, and ICA in lateral recumbency by two independent observers. Measurements were taken from each dog for 20 to 30 min to detect the cervical arterial blood flow velocity. We carried out the standards measurement using the key major descriptors according to Kim et al. [11]. The CCA was identified by placing the transducer in the jugular furrow with the scanning plane directed along the long axis of the neck and at a 45° angle between the parasagittal and dorsal planes. Doppler spectral evaluation was always performed in the longitudinal plane of the artery. The ICA was identified in the same scanning plane as CCA in level ventral to the wing of the atlas, arising at an approximately 30° angle from the ECA and coursing internally. ECA continues in a straight line rostrally from the CCA. Anatomic landmarks in this scanning plane are the mandibular salivary gland and the digastric muscle viewed obliquely as it crosses the caudoventral aspect of the body of the mandible.

Technical parameters for Doppler evaluation of the CCA, ECA, and ICA were set in longitudinal view, where the sample volume was placed in the center of the vessel and the Doppler angle was kept at 45° ± 5. Peak systolic velocity (PSV, centimeters per second), end diastolic velocity (EDV, centimeters per second), and the resistive index (RI) were acquired from the obtained Doppler waveform. The Doppler spectrum was recorded 3 times for all vessels, and the average was calculated as the final value. All measurements were repeated two times with a two-month interval by two observers. On the second occasion, the observers were unaware of the initial values. Practicing measurements were taken under the supervision of a medical specialist in human angiology and cardiology before the study. Training was done on 10 dogs not included in the study.

Arterial spectral Doppler waveforms were evaluated based on key major descriptors and additional modified terms described by the Society for Vascular Medicine and the Society for Vascular Ultrasound. Key major descriptors are presented alongside representative Doppler waveforms [13] and provide:-Arterial waveform nomenclature major descriptors—1. Flow direction (antegrade, retrograde, bidirectional, absent), 2. Phasicity (multiphasic, monophasic), 3. Resistance (high, intermediate, low).-Additional modifier terms:
aUpstroke:
-Rapid: Nearly vertical slope or steep rise to peak systole-Prolonged: Gradual slope to peak systole.
bSharp peak:
-Sharp, single, and well-defined peak, often with maximum-Velocity, within range of the artery being interrogated.
cSpectral broadening:
-Widening of the velocity band in the spectral waveform, a ‘filling in’ of the clear ‘window’ under the systolic peak. Spectral broadening is commonly seen in turbulent flow but can also be seen in the absence of turbulence.
dStaccato:
-A very high-resistance pattern with a short ‘spike’ of velocity acceleration and deceleration followed by a short and low-amplitude diastolic signal reflecting low antegrade flow.
eDampened:
-Combined finding of an abnormal upstroke (delayed) and peak (broad), often with decreased velocity.
fFlow reversal:
-Flow that changes direction, not as part of normal diastolic flow reversal, which may be transient (positional) or consistent with each cardiac cycle (systole/diastole).



### 2.3. Statistical Analysis

All statistical analyses were performed using IBM SPSS Subscription 27 statistical software (IBM Corp^®^., Armonk, NY, USA). Data on body weight, gender, and velocity parameters were collected, and the results were expressed as mean ± standard deviation (SD). Differences were considered significant at *p* < 0.05. A parametric *t*-test was used to compare gender differences. Univariate linear regression analysis was used to determine the correlation between morphometric parameters (such as weight and CCA diameter) and PSV and RI separately for the CCA, ECA, and ICA. Similarly, it was used for the relationship between velocities (PSV and EDV) and RI.

Inter-observer and intra-observer variability was evaluated by means of two-way random single-measure intra-class correlation coefficients for absolute agreement (ICC 2.1). Inter-observer agreement was evaluated between observers separately according to breeds for all parameters and for each breed separately. Intra-observer agreement was evaluated for PSV and EDV, irrespective of breed.

The intra-class correlation coefficient (ICC 2.1) ranged from 0 (no agreement) to 1 (perfect agreement). The strength of agreement was interpreted as follows: <0.5 was considered as poor reliability, values between 0.5 and 0.75 indicated moderate reliability, values between 0.75 and 0.9 indicated good reliability, and values greater than 0.90 indicated excellent reliability.

## 3. Results

### 3.1. Study Population

One hundred and four dogs (51 male, 53 female) between 1 and 5 years old were included in this study. Table 1, Table 2 and Table 3 represent the PSV, EDV, RI and S/D of the CCA, ECA and ICA data measured in the individual breeds.

A strong correlation was observed between the morphometric and hemodynamic parameters (Figure 2). There was a strong negative relationship between the diameter of the CCA vs. PSV CCA and vs. RI CCA (r = −0.52, r = −0.68). A strong negative correlation was also seen between weight vs. PSV CCA and vs. RI CCA (r = −0.54; r = −0.62,) and a strong positive correlation between weight and the CCA diameter (r = 0.87), as well. The relationships between RI and PSV and the EDV of the CCA were derived as follows: Weak for PSV vs. RI (r = 0.28) and strongly negative for EDV vs. RI (r = −0.75) (Figure 3). There was a weak negative correlation between weight vs. RI ECA, vs. PSV ECA and vs. EDV ECA (r = −0.27, r = −0.13, r = −0.19), respectively, and between RI ECA vs. PSV ECA and vs. EDV ECA (r = 0.19, r = −0.28), respectively. Similarly, a weak correlation was observed between weight and PSV ICA and vs. EDV ICA (r = 0.17, r = 0.01), respectively, and RI vs. PSV and vs. EDV (r = 0.26, r = −0.14), respectively. A moderate positive correlation was found between weight and RI ICA (r = 0.37). The relationship of strong correlations is expressed by linear regression in Figure 2 and Figure 3.

### 3.2. Gender and Breed Related Differences

#### 3.2.1. Gender Differences

Gender differences in age, body weight, CCA diameter, intima media thickness (IMT), and hemodynamic variables are shown in Table 4. No significant differences were found between gender for PSV, EDV, and RI for either the CCA, ECA, or ICA. The thickness of the IMT in the CCA and the CCA diameter were significantly higher in males than in females (*p* < 0.05, *p* < 0.001, respectively).

#### 3.2.2. Breed Differences

Significant breed differences were observed in the flow pattern and velocities for PSV and EDV in the CCA, ECA, and ICA.

### 3.3. Evaluation of the CCA

The common carotid flow pattern shows intermediate to high resistance, multiphasic and antegrade waveform with no, minimal, or severe reverse flow. Flow is indicated by sharp systolic peaks, broader than in arteries of a high-resistance flow pattern (Figure 4). A remarkably different morphology in the CCA was observed between groups of dogs, which included differences mainly in the length of diastolic flow and the presence of reverse flow during diastole. Breed-related significant differences for PSV, EDV, and RI of the CCA data are shown in Table 5.

### 3.4. Evaluation of the ECA

Doppler spectral waveform of the ECA shows a high-resistance multiphasic antegrade flow pattern without a reverse flow phase in late systole or early diastole (Figure 5). No significant differences were observed in PSV and EDV ECA among the breeds. RI data from the ECA shows the highest differences between small and giant breeds (*p* < 0.001; *p* < 0.001) and among the giant breeds within their group (*p* < 0.0001) (Table 6).

### 3.5. Evaluation of the ICA

The Doppler waveforms morphology of the ICA shows monophasic low-resistance antegrade flow without reverse flow and relatively high diastolic velocities throughout the cardiac cycle (Figure 6). The ICA waveform displays a more blunted systolic peak and greater diastolic flow than is seen in the ECA waveform. A statistical difference was noted for the RI ICA between the Dachshund and the Hungarian Short-Haired Pointer, Borzoi and Great Dane (*p* < 0.01, *p* < 0.01, *p* < 0.01, respectively). Assessment of the PSV and EDV of the ICA shows maximal variation between large and giant breeds (*p* < 0.001, *p* < 0.0001, respectively) (Table 7).

#### Inter-, Intra-Observer Reliability (ICC)

To assess the inter-observer reliability (ICC_inter_) of PSV and EDV for the CCA, ECA and ICA measurements, agreement between two observers using the same methods was determined.

Excellent ICC_inter_ was detected in 9 (18.75%) measurements, good ICC_inter_ in 30 (62.5%) measurements, and moderate in 9 (18.75%) measurements. In assessed PSV and EDV among breeds, ICC_inter_ was excellent in Borzoi (0.91) and Great Dane (0.91), good in Miniature Schnauzer (0.89), Bavarian Mountain Scent Hound (0.83), Hungarian Short-Haired Pointer and Standard Poodle (0.77), and moderate in Dachshund and Slovakian Hound. Evaluation of the intra-observer reliability (ICC_intra_) of measurements between two repeated measurements by each observer was examined. The ICC of the first observer was excellent for 4 (32.26%) measurements and good for 2 (54.85%). Measurements of the second observer were excellent for 5 (32.26%) and good for 1 (54.85%) (Table 8).

## 4. Discussion

For extracranial carotid examinations, emphasis is placed on PSV, EDV, and RI parameters to assess flow and pathology [3,4,10,12,17,25]. The limiting factor observed in the veterinary literature is breed variation [1,2,4]. Based on the authors’ knowledge, only two reports have described the CCA in healthy Labrador retrievers, and beagles and no Doppler reference values for the CCA, ECA, and ICA have been published in the relevant literature yet [1,4], and they focused on estimating normal blood flow velocity parameters and impedance indexes for the CCA. PSV and EDV are the most accepted Doppler parameters for grading carotid stenoses [10]. Additionally, they are associated with hypertension, brain disorders, age-related changes in vessel wall compliance, and increased flow to supply a collateral pathway [33,34,35,36,37,38]. Data reported in young beagles [4] with PSV 115 ± 17 cm/s and EDV 39 ± 7 cm/s are similar compared to the small/medium breed group described here. On the other hand, values published in Labrador retrievers (PSV 75.8 ± 16 cm/s and EDV 12.2 ± 4 cm/s) [1], as a breed similar to our large breed group, were lower compared to ours (PSV 95.45 cm/s and EDV 26.86 cm/s). For accurate velocity measurements, it is necessary to use an appropriate Doppler angle since it affects the velocity flow proportionally [10,12,17,25,29]. A previous study showed an overestimation of PSV when using a 60° angle, and a Doppler angle of 45° ± 5 was considered to be the most effective for error reduction, therefore, that angle was used in this study [29]. The term “sample volume” defines the three-dimensional area in which the Doppler frequency shifts are measured on the Doppler spectral graph, and subtle changes with sample volume placement and the appropriate Doppler angle may also have affected the velocity measurement variation [17].

The different methodology used in Svicero’s study [1] can perhaps explain the varied results between the studies. By convention, a sample volume size of 2–3 mm and keeping the Doppler angle at 45 ± 5 is considered optimal for minimizing unwanted data. Therefore, we suggest conducting a further study regarding these optimal Doppler parameters in Labrador retrievers and breeds of a similar size.

There was a strong negative correlation of the PSV CCA with the CCA diameter (r = −0.52) and weight (r = −0.54) in breeds. Interestingly, in the literature on human reports, differences in body weight and height have influenced arterial hemodynamics in the carotid artery, particularly blood flow velocities, systolic blood pressure, pulse pressure, wave reflection, and pulse wave velocity [9,32,35,38,39,40]. These findings, in conjunction with the correlation of body weight and velocity parameters found in this study, may support the theory of decreased PSV and EDV due to body weight in large and giant breeds. Moreover, an increased carotid diameter has been reported in the male sex proportionally with height and weight in human medicine [9]. In our study, the diameter of the CCA also strongly correlated with weight (r = 0.87), which was higher in males in our study (*p* < 0.05). These findings are highly consistent with previously presented values in human medicine [9].

The resistance index (RI) is an index of vascular resistance and reflects the global vascular resistance of the downstream territory. RI assesses the ratio of the upstroke of the systolic wave to the end diastolic flow rate: RI = PSV-EDV/PSV [4,13,32]. A weak correlation was found between RI and PSV of the CCA (r = 0.28) and a strong negative correlation with EDV (r = −0.75). It follows from the data that the higher PSV (*p* < 0.01) and lower EDV (*p* < 0.01) in small and medium breeds lead to significantly higher RI (*p* < 0.01) values in comparison with large and giant breeds. The ICA shows the properties of a low-resistance waveform with a more blunted systolic peak and a greater and relatively high diastolic flow and constant resistive index [17,20]. In contrast, the ICA of small and medium breeds had lower PSV (*p* < 0.05) and higher EDV (*p* < 0.001), which indicates a lower RI (*p* < 0.01) compared to large and giant dogs. When we take into consideration the weight and the positive correlation found between the weight and velocity, we can conclude that the weight of dogs has a great influence on the values of the RI index. Currently, we are not able to explain this finding. Presumably this interaction could be dependent on left ventricular mass index, left atrial volume index, and right ventricular systolic pressure, which are most likely breed-dependent variabilities. A similar correlation between weight and RI was observed in studies focused on *arteria testicularis*, which is a low-resistance type of artery. The results are in line with those previously reported on *a. testicularis* in dogs with different weightss and body size, in which a higher RI index was related to a higher weight [8].

The Doppler waveforms morphology of common, external, and internal carotid arteries flow represents a specific character of the vascular bed being supplied [11,32]. The most prominent diversity in the morphology of the spectral waveform was for the CCA between breeds in our study. The common carotid artery flow profile represents an amalgamation of the internal carotid artery and the external carotid artery, and it is assumed to have a relatively low to intermediate resistance character [11,13,17,20]. This may be explained by the preponderance of carotid flow entering the internal carotid artery (≈80%), with sharp systolic peaks and forward flow in diastole with moderate, minimal, or without reverse flow. Our sample of small and medium breeds had the morphology of a spectral waveform with a sharp systolic upstroke, a brisk downstroke, the visible presence of an end systolic notch, and long continuous forward flow throughout diastole that is above the zero-flow baseline (Figure 4). The young beagles included in Lee’s study [4] reached a weight of 10 kg and are equal to the small breed group presented here, in which reverse flow in early diastole was not observed. Medium and large breeds showed a spectral waveform pattern with the same sharp systolic upstroke, similar to small breeds, without the end systolic notch, however. A small, sharp reverse flow was present in early diastole, and a continuous forward flow was short, in contrast to small breeds (Figure 4), which is in line with previously reported values in a large breed [1]. The spectral waveform of the CCA in giant breeds showed a flow pattern with an extension of the reverse blunted flow phase during the first half of diastole, while the rest of the forward flow phase was above the basal line in diastole. Minor variations in the spectral waveform of the ECA were recorded in the length of the diastolic flow, which gradually shortened in large and giant breeds (Figure 3). The morphology of the ICA in the group of small dogs showed a relatively high, sharp peak during systole with a gradually decreasing forward flow in diastole. Medium and large breeds had a similar flow pattern to the ICA, with a sharp drop in the systolic peak and a high diastolic flow. The spectral waveform in giant breeds, especially Great Dane, showed a more pronounced difference between systolic and diastolic flow in the ICA compared to large (*p* < 0.01), medium (*p* < 0.05), and small breeds (*p* < 0.05). Despite the values of the S/D ratio (3.01 ± 0.48) and significant differences in the velocity values, the ICA spectral waveform has a pattern of low resistance flow (Figure 4).

Degenerative and proliferative arterial diseases in dogs are age or disease-related [2,15]. Hyperlipidemia is one of the main risk factors for atherosclerosis in human beings, and Miniature Schnauzer and Shetland sheepdogs have been shown to be at increased risk for primary hyperlipidemia [14,16]. Intima-media thickness (IMT) is widely used as one of the evaluation parameters for arterial wall disease [12,17,20]. Multicenter European, Japanese and Australian studies in people have evaluated the gender-related differences in carotid IMT in a young-to-middle-aged healthy population without carotid disease. This multicenter study showed strongly diversified common carotid artery IMT as a possible consequence of the effect of sex steroids, blood pressure, and anthropometric influences [35,36,40,41]. Although it was not a subject of this study, our findings are in line with previous findings from human medicine, where men had significantly higher IMT compared to women, and it would be interesting to observe the influence of the same parameters in a dog cohort in further studies.

The differences in results obtained by Doppler ultrasound are influenced by many factors, such as the angle of insonation, the width and the placement of the gate, aliasing, error arising from the patient, and error due to the equipment [17,26,27,29,31]. The other possible factor influencing the results is inter-observer variability. However, the above-mentioned measurement errors can also affect the agreement of the observers [28,29,30]. Moreover, results of multicentric studies obtained with different type of equipment can equally affect the results. In this study, all the measurements were made with the same unit. Obesity may also be a source of error, as obese dogs are more difficult to examine. Finally, there might also be variation between dogs in the detectability of neck vessels and the level of dogs’ cooperation. Measuring the velocity in the neck area can be difficult and potentially influence measurements due to subcutaneous fat in dogs and variation in size and anatomy of vessels, mainly the ECA and ICA, which could explain the lowest inter-observer agreement in the ECA and ICA. Cooperative training can reduce the inter-observer variability and bring reproducibility of Doppler measurements to an acceptable level. Previous studies indicate higher agreement and reproducibility with agreeing on a consensus interpretation of the methods, such as studies with a heterogenous group of radiologists [27,29,30].

## 5. Conclusions

This study was conducted with the objective of describing the sonographic appearance and assessment of the hemodynamics of extracranial arteries in healthy canines according to their weight and size. The weight, diameter, PSV, and EDV of the examined dogs had a highly significant effect on the value of the RI index in the common carotid arteries. ECA from all carotid arteries in this study showed the smallest differences in the spectral waveforms and no significant differences in PSV and EDV between breeds. RI, as a resistance evaluation parameter, showed significant differences in ICA and ECA between breeds, which are most likely caused by the weight of the dogs.

These values can be used as reference values for the application of spectral Doppler on the canine carotid arteries. Knowledge of normal Doppler signs of each blood vessel is crucial in their identification.

The present study is the first step for future investigations that will attempt to define this point and compare such results.

## Figures and Tables

**Figure 1 animals-13-01584-f001:**
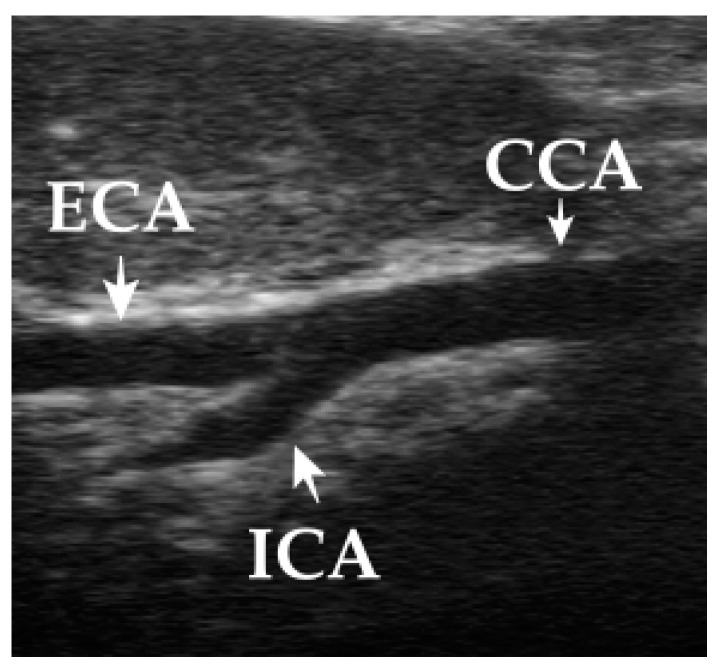
Scanning the carotid arteries by placing the transducer in the jugular furrow with the scanning plane directed along the long axis of the neck and at a 45° angle between the parasagittal and dorsal planes. The carotid bifurcation: CCA—common carotid artery, ECA—external carotid artery, ICA—internal carotid artery.

**Figure 2 animals-13-01584-f002:**
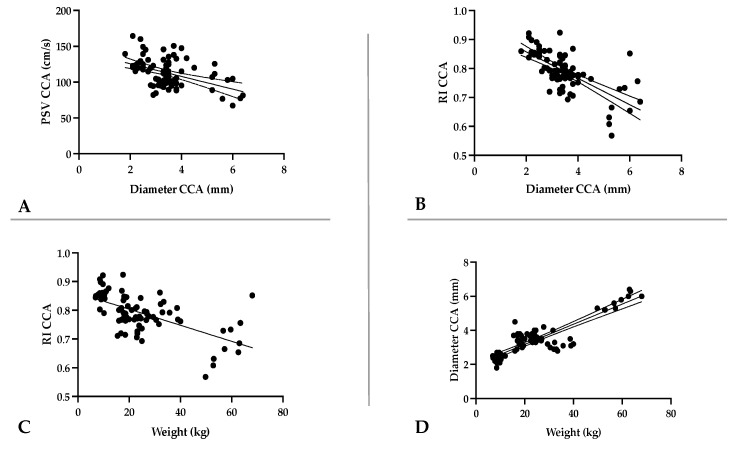
Relationship of the diameter of the common carotid artery (CCA) (mm) with peak systolic velocity (cm/s) in the CCA (**A**); with the resistive index of the CCA (**B**). Relationship of weight with the resistive index of the CCA (**C**); with the diameter of the CCA (mm) (**D**). CCA—common carotid artery; PSV—peak systolic velocity; EDV—end diastolic velocity; RI—resistive index.

**Figure 3 animals-13-01584-f003:**
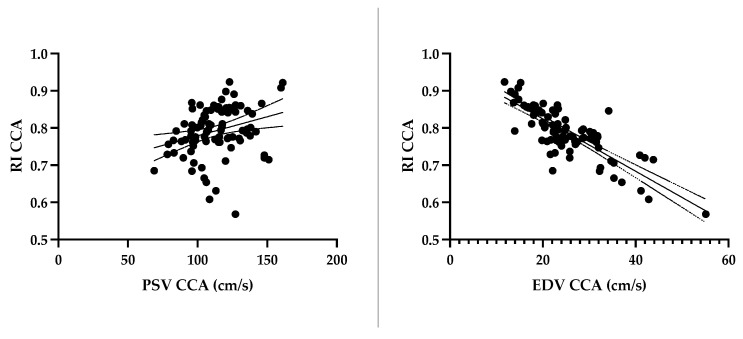
Effect of peak systolic and end diastolic velocity on the resistive index in the common carotid artery. CCA—common carotid artery; PSV—peak systolic velocity; EDV—end diastolic velocity; RI—resistive index.

**Figure 4 animals-13-01584-f004:**
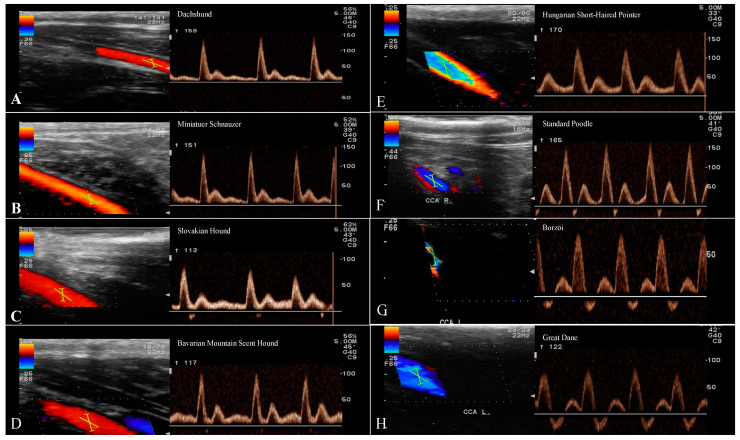
Spectral waveforms of the CCA in different breeds. Spectral waveforms of the CCA in different breeds. Recorded waveforms in small breeds show antegrade, monophasic flow with a sharp systolic upstroke and forward diastolic flow. In medium and large breeds antegrade was observed, but multiphasic flow was represented with minimal reverse flow during early diastole. Giant breeds had a significant reverse flow that created half of the diastolic flow. Key: (**A**)-Dachshund, (**B**)-Miniature Schnauzer, (**C**)-Slovakian Hound, (**D**)-Bavarian Mountain Scent Hound, (**E**)-Hungarian Short-Haired Pointer, (**F**)-Standard Poodle, (**G**)-Borzoi, (**H**)-Great Dane.

**Figure 5 animals-13-01584-f005:**
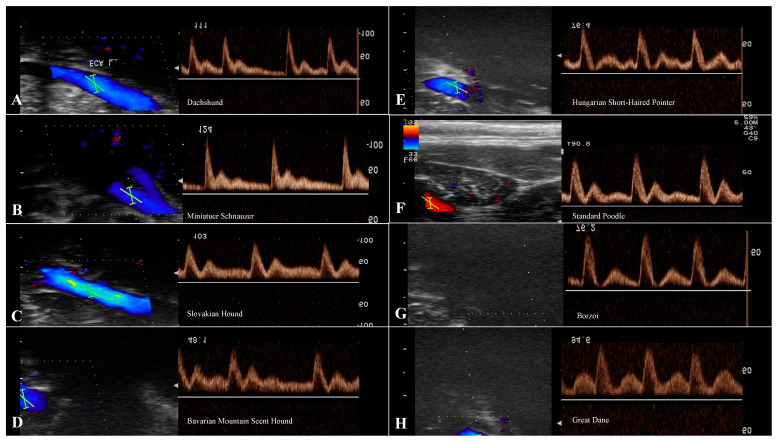
Spectral waveforms of the ECA in different breeds. The representative flow patterns for ECA as a high-resistance artery are antegrade, monophasic flow with a rapid systolic upstroke and forward flow throughout diastole. There is also a clear spectral broadening under the systolic peak. In a large and giant breeds the diastolic flow is shorter like in small and medium breeds. Key: (**A**)-Dachshund, (**B**)-Miniature Schnauzer, (**C**)-Slovakian Hound, (**D**)-Bavarian Mountain Scent Hound, (**E**)-Hungarian Short-Haired Pointer, (**F**)-Standard Poodle, (**G**)-Borzoi, (**H**)-Great Dane.

**Figure 6 animals-13-01584-f006:**
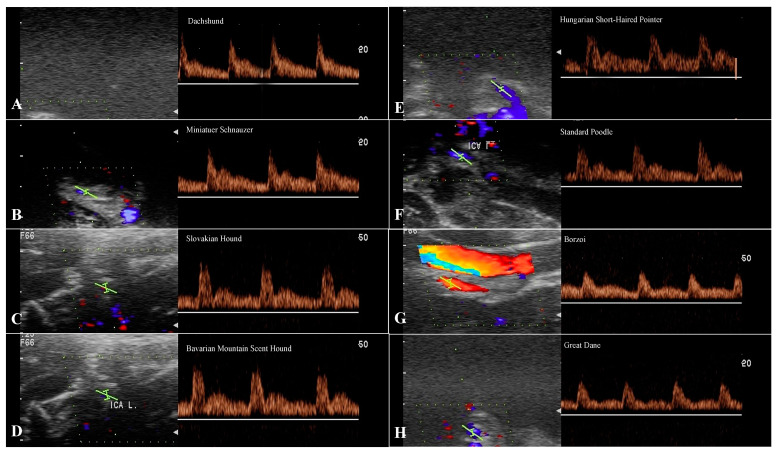
Spectral waveforms of the ICA in the breeds. This is a typical low-resistance antegrade, monophasic arterial flow pattern represented by a relatively rapid systolic upstroke, continued forward flow throughout diastole, and relatively high diastolic velocities. The giant breeds showed a greater difference between systolic peak and diastolic flow than other breeds. Key: (**A**)-Dachshund, (**B**)-Miniature Schnauzer, (**C**)-Slovakian Hound, (**D**)-Bavarian Mountain Scent Hound, (**E**)-Hungarian Short-Haired Pointer, (**F**)-Standard Poodle, (**G**)-Borzoi, (**H**)-Great Dane.

**Table 1 animals-13-01584-t001:** Mean values, standard deviation (±SD), the diameter and hemodynamics parameters of the CCA in all dogs divided by breeds.

		CCA					
		n	Ø (mm)	RI	PSV (cm/s)	EDV (cm/s)	S/D
Small breeds	Dachshund	13	2.5	0.84	130.6 ± 15.56	17.98 ± 4.48	7.84 ± 2.1
	Miniature Schnauzer	13	2.3	0.84	127.3 ± 13.49	19.57 ± 3.68	6.85 ± 1.7
Medium breeds	Slovakian Hound	13	3.5	0.81	108.1 ± 17.39	19.96 ± 6.83	6.47 ± 2.9
	Bavarian Mountain Scent Hound	13	3.6	0.76	115.4 ± 21.7	30.71 ± 7.1	3.88 ± 0.4
Large breeds	Hungarian Short-Haired Pointer	13	3.5	0.76	119.9 ± 16.07	27.52 ± 5.63	4.63 ± 0.98
	Standard Poodle	13	3.5	0.79	106.7 ± 9.2	22.31 ± 4.49	5 ± 0.77
Giant breeds	Borzoi	13	3.2	0.78	96.54 ± 6.21	22.3 ± 1.56	4.49 ± 0.54
	Great Dane	13	5.7	0.66	94.37 ± 18.7	31.43 ± 12	3.40 ± 0.95

CCA—common carotid artery; n—number of dogs; Ø—diameter of CCA; PSV—peak systolic velocity; EDV—end diastolic velocity; RI—resistive index; S/D—systolic/diastolic ratio.

**Table 2 animals-13-01584-t002:** Mean values (± SD) of the hemodynamics parameters of the ECA in all breeds.

		ECA				
		n	RI	PSV (cm/s)	EDV (cm/s)	S/D
Small breeds	Dachshund	13	0.79	76.21 ± 13.77	17.71 ± 4.91	4.77 ± 1.34
	Miniature Schnauzer	13	0.79	78.81 ± 19.23	18.05 ± 6.02	4.76 ± 1.17
Medium breeds	Slovakian Hound	13	0.74	73.08 ± 13.76	19.78 ± 5.29	3.89 ± 0.92
	Bavarian Mountain Scent Hound	13	0.71	80.45 ± 17.78	24.21 ± 7.91	3.57 ± 0.87
Large breeds	Hungarian Short-Haired Pointer	13	0.73	71.74 ± 15.42	19.14 ± 2.58	3.86 ± 0.79
	Standard Poodle	13	0.73	65.98 ± 15.01	20.42 ± 5.62	3.34 ± 0.54
Giant breeds	Borzoi	13	0.79	76.13 ± 4.53	17.14 ± 1.23	4.55 ± 0.39
	Great Dane	13	0.65	68.19 ± 10.15	22.6 ± 5.09	3.12 ± 0.59

ECA—external carotid artery; n—number of dogs; PSV—peak systolic velocity; EDV—end diastolic velocity; RI—resistive index; S/D—systolic/diastolic ratio.

**Table 3 animals-13-01584-t003:** Mean values (± SD) of the hemodynamics parameters of the ICA in all breeds.

		ICA				
		n	RI	PSV (cm/s)	EDV (cm/s)	S/D
Small breeds	Dachshund	13	0.56	29.52 ± 8.86	12.82 ± 4.52	2.40 ± 0.47
	Miniature Schnauzer	13	0.60	32.75 ± 8.47	12.29 ± 1.99	2.66 ± 0.44
Medium breeds	Slovakian Hound	13	0.60	28.55 ± 7.87	11.51 ± 3.43	2.56 ± 0.33
	Bavarian Mountain Scent Hound	13	0.58	31.99 ±16.22	11.54 ± 2.61	2.7 ± 0.74
Large breeds	Hungarian Short-Haired Pointer	13	0.66	24.55 ± 7.19	8.84 ± 3.58	2.98 ± 0.49
	Standard Poodle	13	0.64	24.42 ± 7.71	9.31 ± 3.43	2.73 ± 0.25
Giant breeds	Borzoi	13	0.64	43.26 ± 12.3	17.2 ± 3.18	2.55 ± 0.36
	Great Dane	13	0.66	29.27 ± 8.8	10.17 ± 4.08	3.01 ± 0.48

ICA—internal carotid artery; n—number of dogs; PSV—peak systolic velocity; EDV—end diastolic velocity; RI—resistive index; S/D—systolic/diastolic ratio.

**Table 4 animals-13-01584-t004:** Baseline characteristics of the study population and their hemodynamic parameters.

Variable	Male (n = 51)	Female (n = 53)	*p* < 0.05
Age (years)	2.45 ± 0.98	2.6 ± 1.1	ns
Weight (kg)	26.46 ± 17.14	21.8 ± 13.18	ns
Diameter CCA (mm)	3.56 ± 1.09	3.25 ± 0.92	0.001
Intima media CCA (mm)	0.32 ± 0.04	0.29 ± 0.05	0.05
RI CCA	0.78 ± 0.07	0.8 ± 0.06	ns
PSV CCA (cm/s)	113.3 ± 21.6	112.5 ± 16.5	ns
EDV CCA (cm/s)	24.9 ± 8.4	23.2 ± 6.72	ns
RI ECA (cm/s)	0.74 ± 0.07	0.73 ± 0.06	ns
PSV ECA (cm/s)	76.42 ± 14.32	72.04 ± 12.57	ns
EDV ECA (cm/s)	20.34 ± 5.31	19.57 ± 4.86	ns
RI ICA (cm/s)	0.63 ± 0.07	0.62 ± 0.06	ns
PSV ICA (cm/s)	31.66 ± 12.35	29.18 ± 8.85	ns
EDV ICA (cm/s)	12.05 ± 4.23	11.26 ± 3.86	ns

n—number of dogs; RI—resistive index; PSV—peak systolic velocity; EDV—end diastolic velocity; CCA—common carotid artery; ECA—external carotid artery; ICA—internal carotid artery; statistic significant: *p* < 0.05, *p* < 0.001, ns—non-significant.

**Table 5 animals-13-01584-t005:** Breed-related differences in the CCA.

Breed-Related Differences in PSV, EDV and RI CCA Data			
Breed	PSV CCA *p* < 0.05	EDV CCA *p* < 0.05	RI CCA *p* < 0.05
Dachshund vs. Standard Poodle	0.05	ns	ns
Dachshund vs. Bavarian Mountain Scent Hound	ns	0.001	0.001
Dachshund vs. Hungarian Short-Haired Pointer	ns	0.01	0.001
Dachshund vs. Borzoi	0.01	ns	0.01
Dachshund vs. Great Dane	0.05	0.0001	0.001
Miniature Schnauzer vs. Hungarian Short-Haired Pointer	ns	0.01	ns
Miniature Schnauzer vs. Standard Poodle	0.01	ns	ns
Miniature Schnauzer vs. Borzoi	0.01	ns	ns
Miniature Schnauzer vs. Great Dane	0.05	0.01	0.05
Slovakian Hound vs. Bavarian Mountain Scent Hound	ns	0.01	ns
Slovakian Hound vs. Great Dane	ns	0.01	0.05
Hungarian Short-Haired Pointer vs. Borzoi	0.05	ns	ns
Hungarian Short-Haired Pointer vs. Great Dane	ns	ns	0.05
Standard Poodle vs. Great Dane	ns	0.05	0.05
Borzoi vs. Great Dane	ns	0.05	0.05

CCA—common carotid artery; PSV—peak systolic velocity, EDV—end diastolic velocity; RI—resistive index; *p* < 0.05, *p* < 0.001, *p* < 0.0001, *p* < 0.00001 ns—non-significant.

**Table 6 animals-13-01584-t006:** Breed-related differences in the ECA.

Breed-Related Differences in RI ECA Data		
Breed	RI ECA	*p* < 0.05
Dachshund vs. Great Dane	0.77 ± 0.06 vs. 0.66 ± 0.05	0.001
Miniature Schnauzer vs. Great Dane	0.77 ± 0.06 vs. 0.66 ± 0.05	0.001
Bavarian Mountain Scent Hound vs. Borzoi	0.70 ± 0.08 vs. 0.80 ± 0.13	0.01
Standard Poodle vs. Borzoi	0.70 ± 0.04 vs. 0.80 ± 0.13	0.05
Borzoi vs. Great Dane	0.80 ± 0.13 vs. 0.66 ± 0.05	<0.0001

ECA—External carotid artery; RI—Resistive index; *p* < 0.05, *p* < 0.001, *p* < 0.0001, *p* < 0.00001.

**Table 7 animals-13-01584-t007:** Breed-related differences in the ICA.

Breed-Related Differences in PSV, EDV, and RI ICA Data			
	PSV ICA	EDV ICA	RI ICA
Breed	*p* < 0.05	*p* < 0.05	*p* < 0.05
Dachshund vs. Hungarian Short-Haired Pointer	ns	ns	0.01
Dachshund vs. Borzoi	0.05	ns	0.01
Dachshund vs. Great Dane	ns	ns	0.01
Miniature Schnauzer vs. Borzoi	ns	0.045	ns
Slovakian Hound vs. Borzoi	0.05	0.0105	ns
Bavarian Mountain Hound vs. Borzoi	ns	0.01	ns
Hungarian Short-Haired Pointer vs. Borzoi	0.001	<0.0001	ns
Standard Poodle vs. Borzoi	0.01	<0.0001	ns
Borzoi vs. Great Dane	0.05	0.001	ns

ICA—internal carotid artery; PSV—peak systolic velocity, EDV—end diastolic velocity; RI—resistive index; *p* < 0.05, *p* < 0.001, *p* < 0.0001, *p* < 0.00001 ns—non-significant.

**Table 8 animals-13-01584-t008:** Intra-class correlation coefficient for inter- and intra-observer agreement.

		ICC Inter-Observer					
		PSV			EDV		
		CCA	ECA	ICA	CCA	ECA	ICA
Small breeds	Dachshund	0.75	0.68	0.76	0.79	0.69	0.79
	Miniature Schnauzer	0.85	0.79	0.92	0.88	0.8	0.91
Medium breeds	Slovakian Hound	0.62	0.57	0.63	0.65	0.61	0.62
	Bavarian Mountain Scent Hound	0.86	0.81	0.79	0.83	0.83	0.81
Large breeds	Hungarian Short-Haired Pointer	0.88	0.87	0.85	0.76	0.83	0.8
	Standard Poodle	0.76	0.77	0.8	0.79	0.69	0.8
Giant breeds	Borzoi	0.91	0.86	0.9	0.9	0.89	0.92
	Great Dane	0.93	0.89	0.91	0.89	0.92	0.9
		ICC Intra-Observer					
		PSV			EDV		
		CCA	ECA	ICA	CCA	ECA	ICA
	Observer 1	0.92	0.93	0.87	0.91	0.89	0.93
	Observer 2	0.9	0.89	0.91	0.9	0.92	0.91

ICC—intra-class correlation coefficient; PSV—peak systolic velocity; EDV—end diastolic velocity; CCA—common carotid artery; ECA—external carotid artery; ICA—internal carotid artery.

## Data Availability

The data presented in this study are available on request from the corresponding author.

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
