# Peer review of "Ultrasound Evaluation of Extracranial Cerebral Circulation (The Common, External and Internal Carotid Artery) in Different Breeds of Dogs"

_animals, 2023, doi:10.3390/ani13101584_

Round 1

Reviewer 1 Report

The paper submitted by the authors provides relevant information to previous vascular  literature. Moreover, it is concise and well organized. Before acceptance, I recommend to check the following comments:   Figure 1, please include the plane. Line 123, I understand the advice of a human cardiologist. Nonetheless, I miss the opinion of a board certified veterinary cardiologist. Line 154, many vascular diseases are observed in older animals, why you didn’t include mature dogs in your study? Line 171, are you sure that you didn’t see differences related to age parameters. In humans, important changes can be identified due to. Line 194, this difference could be related to the high differences in weight of animals in this group? Line 354, it would be interesting to explain some reasons for this in the Great Dane since this breed is predisposed to specific valvular diseases. Conclusion section, here I would include some explanation about ECA and ICA. Line 391, ICA is not extracraneal, please modify. Line 469, please include all the authors.

Author Response

Dear Reviewer, first of all, I would like to thank you for your time to review our study and for your comments. Please see the attachment in which I respond to your comments.

Reviewer 2 Report

The manuscript written by Ševčíková et al. is an excellent study of the ultrasound evaluation of the common, external and internal carotid arteries in different breeds of dogs. The study was well-planned and executed.
The number of analyzed individuals in each group is small, but it allows statistical analysis. The number of tables and figures is adequate. The discussion is written in a relevant and interesting way. The conclusions are appropriate and supported by the results. I can only recommend minor technical suggestions that do not affect the quality of work:
-       in the list of authors, there should not be a comma after the Mária Lapšanská  and before the number 1,
-       before “2.1 Study population” should be “2. Material and methods”,
-       location of tables and figures should be closer to where they are cited.
In conclusion, I congratulate the authors on a great job.

Author Response

(The authors gave the same response as above.)

Reviewer 3 Report

Dear Authors,

I reviewed the manuscript entitled"Ultrasound evaluation of extracranial cerebral circulation (the common, external and internal carotid artery) in different breeds of dogs". The topic is very interesting since little is known on carotid artery Doppler evaluation in dogs. The report is well written and clear and can be considered as a starting base for future references.

 I have only minor comments, therefore I recommend minor revision.

Specific comments:

Please change resistance index into resistive index throughout the manuscript

Spectral doppler evaluation: Please add a brief description of ultrasonographic approach to common carotid artery, external and internal carotid artery. 

Line 111: "Measurements were taken from each dog for 20 to 30 minutes to detect the cervical arterial blood flow velocity at rest in lateral recumbency". The sentence is unclear. Did the dog rest in lateral recumbency for 30 minutes?

Lines 124-125:"Arterial spectral Doppler waveforms were evaluated based on key major descriptors and additional modified terms..." Please add a brief description or a figure explaining the mentioned terms described by the Society for Vascular Medicine and the Society for Vascular Ultrasound (Upstroke, Sharp peak; Spectral broadening; Staccato; Dampened; Flow reversal) 

Line 360: Hyperlipidemia is one of the main risk factors of atherosclerosis in human beings. Please add a reference.

Figure 4-5-6: please add a brief description of the spectral waveforms features in different breeds.

Author Response

(The authors gave the same response as above.)
